# Uniaxial compression of calcite single crystals at room temperature: insights into twinning activation and development

Camille Parlangeau[1,2], Alexandre Dimanov[1], Olivier Lacombe[2], Simon Hallais[1], Jean-Marc Daniel[3]

[1]Laboratoire de Mécanique des Solides (LMS), Ecole Polytechnique, F-91128 Palaiseau, France
[2]Sorbonne Université, CNRS-INSU, Institut des Sciences de la Terre de Paris, ISTeP UMR 7193, F-75005 Paris, France
[3]Ifremer, F-29280 Plouzané, France

*Correspondence to*: Camille Parlangeau (camille.parlangeau@gmail.com)

**Abstract.**

E-twinning is a common plastic deformation mechanism in calcite deformed at low temperature. Strain rate, temperature and confining
pressure have negligible effects on twinning activation that is mainly dependent on differential stress. The critical resolved shear stress (CRSS) required for twinning activation is dependent on grain size and strain hardening. This CRSS value may obey the Hall-Petch relation, but due to sparse experimental data its actual evolution with grain size and strain still remains a matter of debate.

In order to provide additional constraints on twinning activation and development, new mechanical tests were carried out at room temperature on unconfined single crystals of calcite, with different sizes and crystallographic orientations. Uniaxial
deformation was performed at controlled displacement rate, while the sample surface was monitored using optical microscopy and high resolution CCD camera. The retrieved macroscopic stress-strain behaviour of the crystals was correlated with the surface observations of the deformation process.

Results show (1) the onset of crystal plasticity with the activation of the first isolated mechanical twins during the strain hardening stage, and (2) the densification and thickening of twin lamellae during the steady state flow stress stage. Such
thickening of twin lamellae at room temperature emphasizes that calcite twin morphology is not only controlled by temperature. The different values for the CRSS obtained for the activation of isolated twins and for the onset of twin densification and thickening questions the appropriate value to be considered when using calcite twin data for stress inversion purposes.

## 1 Introduction

Calcite is a common mineral in the earth upper crust, forming different types of sedimentary (carbonates) or metamorphic (marbles) rocks. Deformation modes of calcite aggregates have been investigated experimentally since the early 50's (e.g., Turner, 1949; Griggs and Miller, 1951; Handin and Griggs, 1951; Turner and Ch'ih, 1951; Griggs et al., 1951, 1953; Friedman and Heard, 1974; Schmid and Paterson, 1977; Rowe and Rutter, 1990; Lacombe and Laurent, 1996; De Bresser and Spiers, 1993, 1997; Laurent et al., 2000; Rybacki et al., 2013). These studies have provided information about the different
mechanisms allowing calcite crystals to accommodate deformation at various pressure (P), temperature (T) and strain rate

conditions. E-twinning is the plastic deformation mechanism that prevails in calcite crystals at low strain and low temperature. Strain rate, temperature and isotropic stress state (confining pressure) have negligible effects on twinning activation that is mainly dependent on differential stress and grain size (Rowe and Rutter, 1990). Temperature has an impact on the thickness of twin lamellae (Burkhard, 1993). Twin lamellae are commonly observed to be thin (≤1 µm) below 170-200°C and to become thicker (2-5 µm) above this temperature for a given amount of twinning strain (e.g., Ferrill et al., 2004). However, this relationship between twin thickness and temperature has been recently challenged by Rybacki et al. (2013) who emphasized that the increasing duration of stress application – hence the increasing strain - may cause widening of twin lamellae even at room temperature. Another matter of debate deals with the existence (and with the value) of a threshold stress for activation of twinning, which is defined as for crystal slip plasticity as a critical resolved shear stress value (CRSS) (Tullis, 1980; Ferrill, 1998). Several authors have tried to constrain this value (Turner et al., 1954; Lacombe and Laurent, 1996; De Bresser and Spiers, 1997; Laurent et al., 2000; Lacombe, 2001, Rocher et al., 2004; Covey-Crump et al., 2017). The CRSS is mainly dependent on grain size and is subjected to strain hardening: its value increases with the amount of accommodated strain and decreases with increasing grain size (Turner et al., 1954; Tullis, 1980; Rowe & Rutter, 1990; Laurent et al., 2000). This CRSS value may therefore obey the Hall-Petch relation, but due to sparse experimental data, the actual evolution of the CRSS with grain size and strain still remains poorly constrained.

To sum up, most mechanical experiments to date were carried out at high temperature in order to investigate gliding systems in calcite. Very few data are available on the plastic behaviour of calcite crystals at low temperature, in spite of the ubiquitous deformed carbonate rocks in sedimentary basins. Hence, new mechanical tests were carried out at room temperature on unconfined single crystals of calcite, with different sizes and crystallographic orientations. Uniaxial deformation was performed at controlled displacement rate, meanwhile the sample surface was monitored using optical microscopy (reflected light) and high resolution CCD (charge-coupled device) camera. The retrieved macroscopic stress-strain behaviour of the crystals was correlated with the surface observations of the deformation process. Special attention was paid to the resolved shear stress value required to activate twinning and to the sequence of activation and thickening of calcite twins.

## 2. Materials and Methods

### 2.1 Materials

In addition to the grain size dependence of twinning, Newman (1994) highlighted the fact that the grain size distribution could have an impact on the CRSS value. This author indicates that a grain surrounded by 10 grains is more easily twinned than the same grain surrounded by only 5 grains. Thus, in order to circumvent this problem, we decided to work on single crystals.

Iceland spaths were first deformed in order to establish the appropriate deformation protocol to be applied in terms of sample preparation, loading rates, and conditions for in the situ optical/SEM monitoring (supplementary material). This preparation phase allowed for instance to select the most suitable CCD cameras and to establish the SEM imaging conditions.

However, because these spaths always contained pre-existing cleavage cracks, twin lamellae and micro-fluid inclusions, they could not be used further (1) to produce several perfectly similar samples with respect to size, crystal orientation and pre-existing defects, and (2) to accurately infer the CRSS value of interest because the grain-scale heterogeneities may have caused stress concentrations and because the initial strain was associated with some amount of work-hardening that could not be quantified precisely. A way to remove this work-hardening effect would have been to heat the samples. De Bresser and Spiers (1997) healed the initial intra-crystalline strain by annealing at a temperature of 500 °C for 24h: healing occurs by formation of sub-grains as a result of progressively stacking of free dislocations at sub-grain boundaries. However, these sub-grain boundaries could have affected the propagation and spreading of twins and caused non uniform stress distribution at the grain scale, thus biasing the results.

The choice was then made to perform uniaxial compression of single crystals of pure synthetic calcite ($CaCO_3$) of optical quality (purchased from SurfaceNet). Working on such synthetic material ensured the initial quality of the deformed single crystals: limited dislocation density, crystals free from fluid inclusions, cleavages and twins. In order to assess the grain size effect, we tested parallelepiped shaped specimens with dimensions of 4x4x8 $mm^3$ and 3x3x6 $mm^3$. Two samples have their long axis along the direction $[2\bar{1}\bar{1}0]$ and two samples along the direction $[01\bar{1}0]$ (Fig. 1), so that the uniaxial stress could be applied along two different crystallographic orientations.

### 2.2 Methods

In the previous studies calcite samples were always observed in post mortem conditions. Since the purpose of this study was to detect the very early activation of twinning and to characterize its subsequent development, the choice was made to continuously monitor the experiments using either optical microscopy (with a high-resolution camera and SYLVIA software, Bornert et al., 2010) or during in situ SEM testing at low vacuum (60-80 Pa) with a miniaturized uniaxial press. Macroscopic deformation is provided by an optical displacement sensor (solartron), measuring the displacement of the mobile piston. This is the first time that characterization of twinning deformation of calcite was performed coupled with in situ microscopy.

All experiments were performed at the constant strain rate of $10^{-5}$ $s^{-1}$, and at room temperature. The compression was applied along the longest axis of the crystals and the corresponding Schmid's factors were calculated for the two differently oriented kinds of single crystal (Fig. 1). When the first twin lamellae appeared, the sample was unloaded and removed from the press. EBSD (electron back-scattering diffraction, FEI Quanta 600, AZtecHKL) analysis was performed in order to determine whether the expected e-twins (with the highest Schmid's factor) have been activated. Afterwards, the single crystal was replaced into the deformation rig and further subjected to uniaxial compression in order to observe the evolution of twinning deformation (twin growth, densification and widening) and the corresponding strain hardening with the increase of the applied axial strain.

## 3. Results and discussion

### 3.1 Results

Three representative experimental results are shown hereinafter. In general, the loading curves present three distinct major stages. Stage I (or phase–in) corresponds to the emplacement of the loading column, that is the alignment of pistons and samples (and possibly edge effects of the loaded faces, due to small departures from parallelism). Stage II represents the pseudo elastic regime, which averages the sample behaviour and the compliance of the machine. It is hence characterized by an apparent Young's modulus, which is substantially lower than expected for calcite ($\approx 1.3$ GPa). Stage III exhibits dominantly the plastic regime, with twinning, but also some minor contribution of micro-cracking. Stage III may be divided in two sub-stages. Stage IIIa lasts about $1 - 1.5$ % shortening and is characterized by pronounced strain hardening, with activation of a few early twins. Stage IIIb lasts 2 - 4 % shortening, corresponding to a continuous densification of twinning. The latter stage is characterized by a strongly serrated loading curve. The serrations fluctuate around a nearly steady state flow stress, establishing very limited strain hardening, or a plateau-like regime.

Figure 2 shows the loading curve of the first experiment carried out on a small sample ($3 \times 3 \times 6$ mm$^3$), compressed along $[01\bar{1}0]$ (configuration 1 in Fig. 1). The specimen was first deformed until 1.4% shortening in stage IIIa. The very first twins were detected at about 0.3% shortening. Twin activity progressively increased during strain hardening, which is indicated by the numerous drops of axial stress (serrations). After reaching 1.4% shortening the specimen was completely unloaded and removed for observations, prior to a second compression test. At this step the prescribed shortening was of about 151 μm. High resolution SEM micrographs were used to measure twin lamellae thickness, which allows us to determine that twinning contributed to this axial displacement by a length of 116 μm. Optically we could see by transparency a few twin lamellae that do not cross the whole crystal up to the surface and this could explain part of the difference between estimated shortening by twinning and the prescribed shortening. The optical error of measurement of the thickness of individual twin lamellae on pictures may have also contributed to the mismatch. During the second loading, the sample rapidly reached the stage IIIb, characterized by nearly steady state flow stress. This "plateau-like" curve is serrated, composed of numerous and regular segments, corresponding to strain hardening stages followed by sudden stress drops. The optical monitoring shows that the latter can be associated with the continuous densification of twinning. The test was stopped at the onset of micro-cracking prior to failure at about 6 % shortening (Fig. 2).

The thickness of the first twin lamellae was about 2-7 μm. At the very beginning of loading, micro-cracking nucleated right at the upper interface between the piston and the single crystal (Fig. 2). The lack of confining pressure probably favoured micro-fracturing. However, 1) the micro-crack was localized very close to the interface, 2) it did not propagate further within the bulk sample during loading. These observations indicate that micro-cracking did not interfere with the overall response of the sample, which remained dominated by twinning plasticity. The second loading phase of the sample led to the establishment of the stage IIIb (Fig. 2), which shows a strong change in the rate of twinning activation, with a period of high densification and also of twin lamella thickening. These events are clearly correlated with the different strain hardening stages and sudden

stress drops observed on the serrated loading curve representing the macroscopic behaviour. Overall thickening of twin lamella occurred in 3 steps: (1) densification of twin lamellae, (2) individual thickening of each existing twin lamella and (3) merging of thickened neighbouring twin lamellae. For the millimetre-sized single crystals compressed up to a few percent at room temperature and pressure, the thickness of a twin lamella can reach 90 µm for the largest ones. During the stage IIIa, the sample was removed from the uniaxial press to determine the twin lamellae crystallographic orientation using EBSD. This orientation was mostly used to calculate the Schmid's factor and the associated applied resolved shear stress and particularly the value at twin activation (CRSS). The results show that for all the samples the expected (Fig. 1) twin lamellae were activated during the experiments. For the present orientation, with loading along $[01\bar{1}0]$, the calculated Schmid's factor is 0.39 and the corresponding CRSS is 0.44 MPa.

The second experiment was carried out on single crystals of both sizes (3 x 3 x 6 mm$^3$ and 4 x 4 x 8 mm$^3$) compressed along $[2\bar{1}\bar{1}0]$ (Fig. 1). For the smaller sample, the macroscopic behaviour is comparable to that described previously, with 3 different stages: (I) emplacement, (II) elastic deformation, (IIIa) plastic yielding and strain hardening flow (blue curve, Fig. 3), (IIIb) macroscopically steady state plastic flow, characterized by a serrated "plateau-like" stress-strain curve. Though, the latter is less regular than the stage IIIb curve in the figure 3. The optical monitoring allowed similar observations as for the previous sample (Fig. 2), aside from the fact that there was no early micro-cracking during the stage IIIa (Fig. 3). The densification of twins during the stage IIIb was not as intense as for the previous experiment (Fig. 3). For this sample orientation, the activation of two equivalent twin systems (i.e., with the same Schmid's factor, Fig. 1) was expected. However, only one system was activated, possibly due to a small disorientation of the crystal lattice of less than 3°. The CRSS calculated using the Schmid's factor at the appearance of the twinned lamella is 1.25 MPa.

The last experiment was carried out on a larger sample (4 x 4 x 8 mm$^3$), but with the same orientation as the previous one (Fig. 1, configuration 2). The macroscopic behaviour is roughly similar to the previous experiment, with emplacement, pseudo elastic and strain hardening stages (Fig.4). But, it also differs in some details. The plastic yielding is more difficult to determine. The gradual slope change observed at about 0.3-0.4% of axial deformation represents the transition from emplacement to pseudo elasticity, without any activation of twinning. The very first twins appeared at about 0.7% of axial deformation and at 4 MPa flow stress. These observations are roughly in agreement with those from the previous test, where twinning appeared at about 0.5 % and 4 MPa sample shortening and strength. But, as opposed to the previous case, we observed the activation of the two distinct (but equivalent) twin systems, whilst for the previous experiment only one system was active, despite the fact that both samples have the same crystallographic orientation with respect to the uniaxial loading (Fig. 1, configuration 2). Another major difference in the macroscopic behaviour is that the stage IIIb does not present any plateau-like stress-strain curve. The latter is also serrated, but as a whole it shows a strain hardening evolution, until extensive micro-cracking developed at about 8 MPa flow stress and led to the sample failure. Optical monitoring and EBSD characterization show the development of the twinning patterns (the two systems are represented in blue and green, Fig. 4) during the different stages. Stage IIIa shows the first twins and limited micro-cracking. Stage IIIb shows the densification and the thickening of both twin systems, but also micro-cracking. Observed strain hardening is probably due to the interactions between the two,

simultaneously active twin systems. The densification and twin lamellae thickening are not as pronounced as in the previous test. The CRSS calculated using the Schmid's factor at the appearance of twinned lamella is 1.24 MPa for both twinned planes.

## 3.2 Discussion

Despite the rather limited number of deformed synthetic crystals (4), our results show qualitatively the same patterns for the loading curves as for the Iceland spaths (supplementary material), which makes us confident about the reproducibility of the results and the conclusions drawn about the effect of orientation and grain size.

### 3.2.1 Macroscopic behaviour of deformed samples

The macroscopic deformation behaviour of all our single crystals exhibits 3 major stages revealed by the stress-strain curves, in agreement with the findings of de Bresser and Spiers (1993, 1997). The initial stage (I) is the period of preloading during which the full contact between pistons and samples is progressively established and is not the same for each specimen, because it depends on the exact sample geometry (parallelism of loaded faces). Eventual stress concentrations would expectedly be more pronounced during this emplacement stage; however the micro-cracks did not appear at this stage. Micro-cracks essentially developed later on, during the pseudo-elastic phase (stage II) and the plastic phase (stage III). This may indicate that micro-cracking is a complementary and necessary deformation mechanism, allowing for accommodation of local incompatibilities of strain, for instance at sample-piston interfaces (full contact, stages II and III). The latter are actually expected, because twinning is not isotropic and the pistons have not enough degrees of freedom to fit the heterogeneous sample strain if solely resulting from twinning. For these reasons, retrieving the real Young modulus was not possible, so only an apparent and much lower modulus than the expected one was estimated for calcite. Stage (II), which corresponds to a pseudo-elastic deformation period, is combining both elastic deformation of the material and of the loading frame (made of aluminium). The plastic yield strength is reached when twinning activation starts. The latter is evidenced by the change in the slope of the stress – strain curve. Beyond this threshold, deformation stage (IIIa) is characterized by strong strain hardening, which is related to increasing twinning activation and slight densification of twin lamellae. Stage (IIIb) is the second phase of twinning-related plastic deformation, with strong increase in twin density and substantial lamellae spreading and thickening. At this last stage, the stress – strain curve is strongly serrated, made of segments of strain hardening followed by sudden stress drops, leading to the formation of a "plateau-like" stress-strain curve. Both stages IIIa and IIIb partly involve micro-cracking as a secondary deformation mechanism, which however remained limited and never propagated throughout the whole specimen before it reached several percents of shortening due to dominant twinning.

### 3.2.2 Relationship between macroscopic behaviour and micro-structures

During stage (II) deformation is essentially elastic, hence there is no development of any specific micro-structure. The following plastic domain is evidenced by the appearance of the first twin lamellae during stage IIIa and by further twin

densification and thickening during stage IIIb. Stage IIIa may also exhibit some minor micro-cracks that are very localized and do not cross throughout the crystal. Such behaviour could have been avoided by applying confining pressure, or by allowing for lateral free displacements of the pistons. The first isolated twin lamellae appeared during stage (IIIa). This early twinning stage is characterized by strain hardening and lasts up to 1 – 1.5 % shortening. The final plastic stage IIIb occurred for further loading. During the deformation at nearly steady state flow stress ("plateau-like" stress-strain curve) two phenomena were observed for all the samples: (1) densification of twin lamellae in bursts, and (2) thickening of twin lamellae in bursts. In general, the twin lamellae that formed near the loaded sample faces did not thicken very much, while the twin lamellae forming at the central portion of the crystals could reach up to 90 μm in thickness. Thickening occurred in the following sequence: (1) densification of twinning, (2) thickening of the individual lamellae and (3) merging of thickened lamellae. The widest twin lamellae are finally the result of several merging events. Some lamellae from the central part of the specimen, emerging at the free lateral surfaces, show spectacular thickening. In contrast, lamellae emerging at the sample/piston interfaces remained thinner. This difference may be due to the boundary conditions, that is to say the absence of confining pressure at the lateral free surfaces that may have favored thickening, whilst the stress state and the frictions at the faces loaded by the pistons may have in contrast inhibited thickening and merging of twin lamellae.

### 3.2.3 Twin Lamella thickness

Twinning is a deformation mechanism that accommodates the applied bulk strain with some twin lamellae, the thickness of which is controlled by temperature (Burkhard, 1993): a given amount of strain is accommodated by a number of thin twins at temperature lower than 170-200°C and by fewer but thicker twins at temperature above (e.g., Ferrill, 1991, 1998). Calcite twin lamellae are usually considered thick when they exceed 2-5 μm. Rybacki et al. (2013) questioned this statement by documenting 3 μm-thick twin lamellae during their experiments at room temperature. Our experiments confirm their findings in that the duration of stress application, hence the resulting strain, has a significant impact on twin lamella thickness even at low temperature. This means that calcite twin morphology (e.g., thickness) cannot be used as a straightforward low temperature deformation geothermometer (Ferrill et al., 2004).

However, the large amount of thickening (up to 90 μm) observed during our experiments is likely due to the lack of confining pressure and/or to the free surfaces of the samples. This means that our results cannot be directly and easily extrapolated to natural aggregates to date. Actually, confining the samples with a fluid, or leaving an unconfined free surface during uniaxial compression could affect twin thickening, with respect to confinement by crystalline grains. In the latter case, within a polycrystal, the crystallographic orientation of the neighbours will greatly condition the amount of crystal shear that can be transmitted across the grain boundaries. The crystallographic constraints, in terms of shear strain compatibility, along the grain boundaries would favour the occurrence of numerous distributed thin twins. Conversely, the presence of fluid confined free surfaces could favor the localized development of thick twins. Following our reasoning, in highly porous carbonate rocks, most of the constitutive calcite grains offer free surfaces to the porous space. In other words, most of the

grains are only partly constrained by neighbouring grains. Stress enhancement at grain-to-grain contacts and the presence of unconstrained surfaces at pore space possibly facilitate activation and development of thick twin lamellae.

### 3.2.4 CRSS value

One goal of this study was to constrain the evolution of the CRSS value as a function of grain size. In commonly encountered reservoir carbonate rocks (particularly oolitic rocks), the mean grain size is usually lower than one millimetre. In this study, the single-crystal samples were necessarily much bigger and the results may possibly not be directly extrapolated to natural aggregates. The evolution of the CRSS value with grain size is suggested to follow the equation of Hall-Petch (Covey-Crump et al., 2017). Figure 5 summarizes the results of previous studies about the evolution of the CRSS value vs grain size. The CRSS values determined during this study plot in the predicted asymptotic part of the curve, so no clear trend can be derived for millimetre-sized grains.

Defining the CRSS for twinning activation is not straightforward, and continuous monitoring of the experiment is required. In this study, two grain sizes have been tested: $3 \times 3 \times 6$ mm$^3$ and $4 \times 4 \times 8$ mm$^3$. These grain sizes are not often encountered in natural rocks. From these analyses, the CRSS value is around $0.90 \pm 0.35$ MPa (Fig. 5). These values are very low compared to previous studies (De Bresser and Spiers, 1997; Turner et al., 1954), except for unconfined samples from Turner et al (1954) (Fig. 5). However, it is worth mentioning that a direct comparison of our results with those from other similar studies is not easy for several reasons:

- Turner et al. (1954) used natural Iceland spaths. We show (supplementary material) that we also used Island spaths for establishing the experimental protocol. The results are not directly comparable with those obtained from the synthetic (optical quality) crystals. As said previously, the natural calcite single crystals are already slightly strained (some twins are visible), or cleaved, and most contain micro-fluid inclusions. All these defects induced strain hardening during the experiments, hence likely increased the determined CRSS values for twinning activation (see Figs. 2, 3 and 4 in the supplementary material). This is the likely reason why, as expected, our synthetic samples provide a CRSS value slightly lower than those of Turner et al. (1954). Overall, using natural samples implies to start from an undefined initial state, which questions the validity of the retrieved CRSS values.

- De Bresser and Spiers (1997) annealed their Iceland spath specimens in order to restore the previous cumulated strain. Unfortunately, as explained earlier, this procedure leads to the formation of sub-grain boundaries, that affect the propagation and spreading of twins and cause non-uniform stress distribution at the grain scale, which is potentially biasing the results.

Moreover, for both of these previous studies, the determination of the CRSS was based on post mortem observations, which precluded accounting for the loading history and probably the detection of the earliest twinning events. Other possible issues deal with the likely influence of the crystal chemistry and the starting defect density that might affect the lattice structure, which need further investigation.

Some questions also arise about the CRSS value determined from the macroscopic behaviour. For instance, one can wonder whether the CRSS value has to be considered at the appearance of the first isolated twin lamella or at the "plateau" marked by densification and thickening of twin lamellae.

Finally, as for twin thickening, another point is to what extent CRSS values determined from single crystals may be extrapolated to an aggregate, should the grain size be homogeneous and similar to that of the single crystals from which the CRSS was determined. Again, the quantitative characteristics of twinning in a single crystal may not be directly and easily transposed to what happens in twinning grains within an aggregate. Indeed, as mentioned earlier, grain boundaries are obstacles to twin propagation and twinning requires additional deformation mechanisms ensuring shear strain compatibilities between adjacent grains, because twinning plasticity is inherently anisotropic. The application of mechanical data gathered on single crystals to an aggregate thus remains an issue. For that purpose, analytical mean field, or computational full field homogenization models must be applied to try predicting calcite aggregate behaviours. This is much beyond the scope of the present work, which essentially gives the basic input for further homogenization modelling.

## 4. Conclusion

The macroscopic behaviour shown on direct observations of single calcite crystals deformed uniaxially at room temperature was tightly correlated with the development of twinning microstructures. The study shows (1) the early onset of crystal plasticity with the activation of the very first isolated mechanical twins during the strain hardening (stage II), (2) the densification and thickening of twin lamellae during the final stage. The latter accounts for most of the irreversible strain and corresponds to episodically fluctuating crystal strength, with strain hardening increments followed by equivalent stress drops related to twinning events.

Thickening of twin lamellae occurs in two steps with increasing deformation: (1) thickening of the individual lamellae and (2) merging of thickened lamellae. Our observations of thickening of twins at low temperature confirm recent results obtained by Rybacki et al. (2013).

The different values of the CRSS (i.e. $0.90 \pm 0.35$ MPa) for the activation of isolated twins and for the onset of densification and thickening in our deformed samples bring questions about the appropriate value to be considered when using calcite twins for inversion purposes aiming at retrieving paleo-stress orientations and magnitudes (e.g., Lacombe, 2007, 2010; Parlangeau et al., 2018). Furthermore, we did not see any clear grain size effect on the CRSS value. One can wonder whether this may reflect the expected asymptotic evolution of the CRSS with large grain size as predicted by the Hall-Petch relation.

Additional occurrence of micro-cracking events suggests either imperfections of sample geometry or the need for complementary and concomitant accommodation mechanisms due to twinning anisotropy. These observations highlight the importance of direct observations during mechanical testing in order to interpret mechanical data.

Finally, at the moment, our results cannot be directly and easily extrapolated to natural aggregates. The application of mechanical data gathered on single crystals to an aggregate is another important topic to be addressed in the future. The

reported preliminary results nevertheless pave the way to more confident investigations of different grain sizes and of both natural and synthetic aggregates.

## Acknowledgments.

This study is part of Camille Parlangeau's PhD work conducted in the Institut des Sciences de la Terre de Paris (ISTeP) of Sorbonne Université and in IFP Energies Nouvelles and supported by IFP Energies Nouvelles. Purchase of synthetic single crystals was made possible through a grant provided by the LABEX Matisse (Sorbonne Université, IFP Energies Nouvelles). The experiments were financially supported by the LMS- Ecole Polytechnique and the Chaire Energies Durables (EDF foundation). Camille Parlangeau warmly thanks E. Kohler and S. Schueller for their help and fruitful advices. The authors thank Hans de Bresser and an anonymous reviewer for their thoughtful comments that helped improve the initial version of the manuscript.

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

— ○ Unfavorable to twinning

De Bresser et Spiers (1993, 1997) ■ Unfavorable to twinning

● plane  ○ direction

| | Schmid Factor | | |
|---|---|---|---|
| | $e_1$ | $e_2$ | $e_3$ |
| **Turner et al. (1954)** | | | |
| 1. Extension // to C | 0,39 | 0,39 | 0,39 |
| 2. Extension // to $e_2$ | 0,42 | 0 | 0,42 |
| 3. Extension // to $r_1$ | 0,003 | 0,32 | 0,003 |
| 4. Comp. in $m_1$ perpendicular to C | -0,22 | 0,44 | -0,22 |
| 5. Extension in $m_1$ perpendicular to C | 0,22 | -0,44 | 0,22 |
| 6. Extension 81° to C; 35.5° to $r_1$ | -0,13 | -0,28 | -0,13 |
| 7. Compression // to $r_1$ | 0 | 0,32 | 0,003 |
| 8. Compression 30° to C; 75° to $r_1$ | 0,40 | -0,07 | 0,40 |
| 9. Compression // to C | -0,39 | -0,39 | -0,39 |
| De Bresser et Spiers (1993, 1997) | 0 | -0,5 | -0,3 |
| **This study** | | | |
| 1. Compression along [01-10] | 0,39 | 0,10 | 0,10 |
| 2. Compression along [2-1-10] | 0 | 0,30 | 0,30 |

**Figure 1: Calcite lattice projection in Schmid's lower hemisphere with the information about the force vector applied used in different studies (Turner et al., 1954; De Bresser and Spiers, 1993, 1997) and in this study. The table summarizes the calculated Schmid's factor for each e-twin plane ($e_1$, $e_2$ and $e_3$) as a function of the applied force orientation. Pink stars correspond to the applied compression orientations used on the synthetic single-crystal of calcite.**

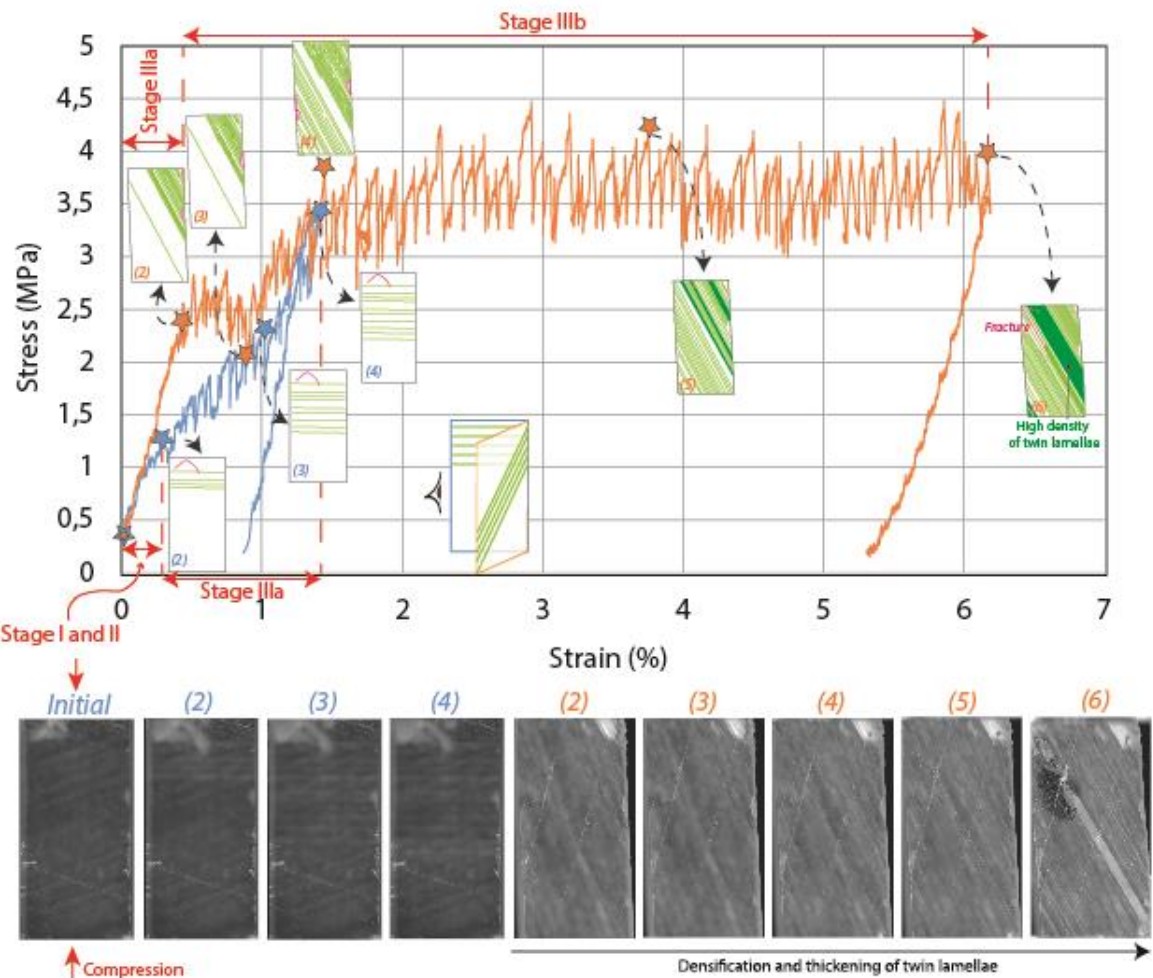

**Figure 2: Results from sample 3 x 3 x 6 mm³ compressed along the direction [01$\bar{1}$0]. Stress (MPa) versus strain (%) curve showing the three stages described in the text. Between the blue and orange curves, the sample has been removed from the press in order to determine the orientation of the first twin lamella. Each star on the curves corresponds to one of the high-resolution pictures below. In the different pictures, the twin lamella are in light green color, the dark green color is used to highlight a high density of twin lamella. Fractures are in pink color.**

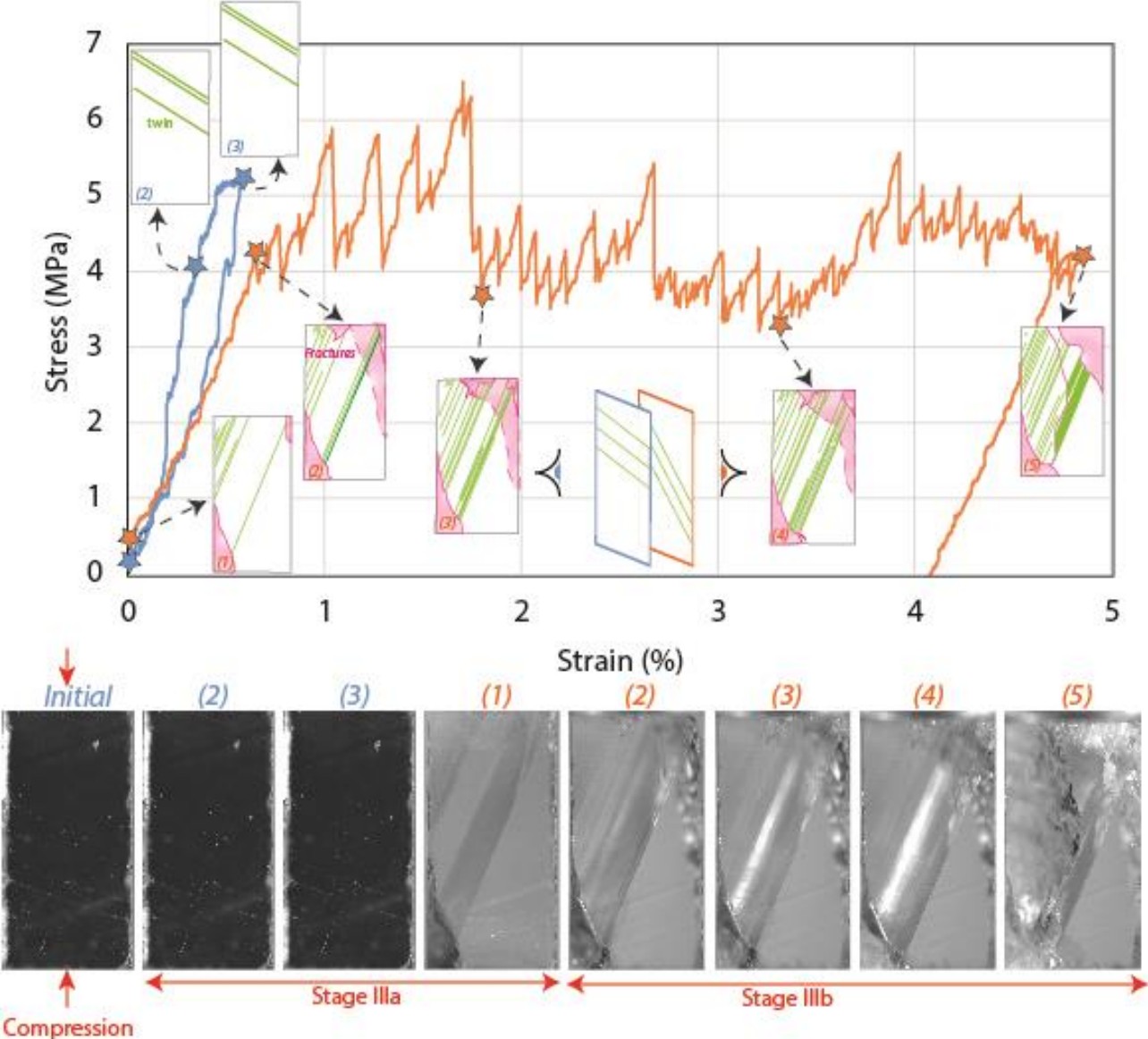

**Figure 3. Results from sample 3 x 3 x 6 mm³ compressed along the direction $[2\bar{1}\bar{1}0]$. Same key as in Fig.2.**

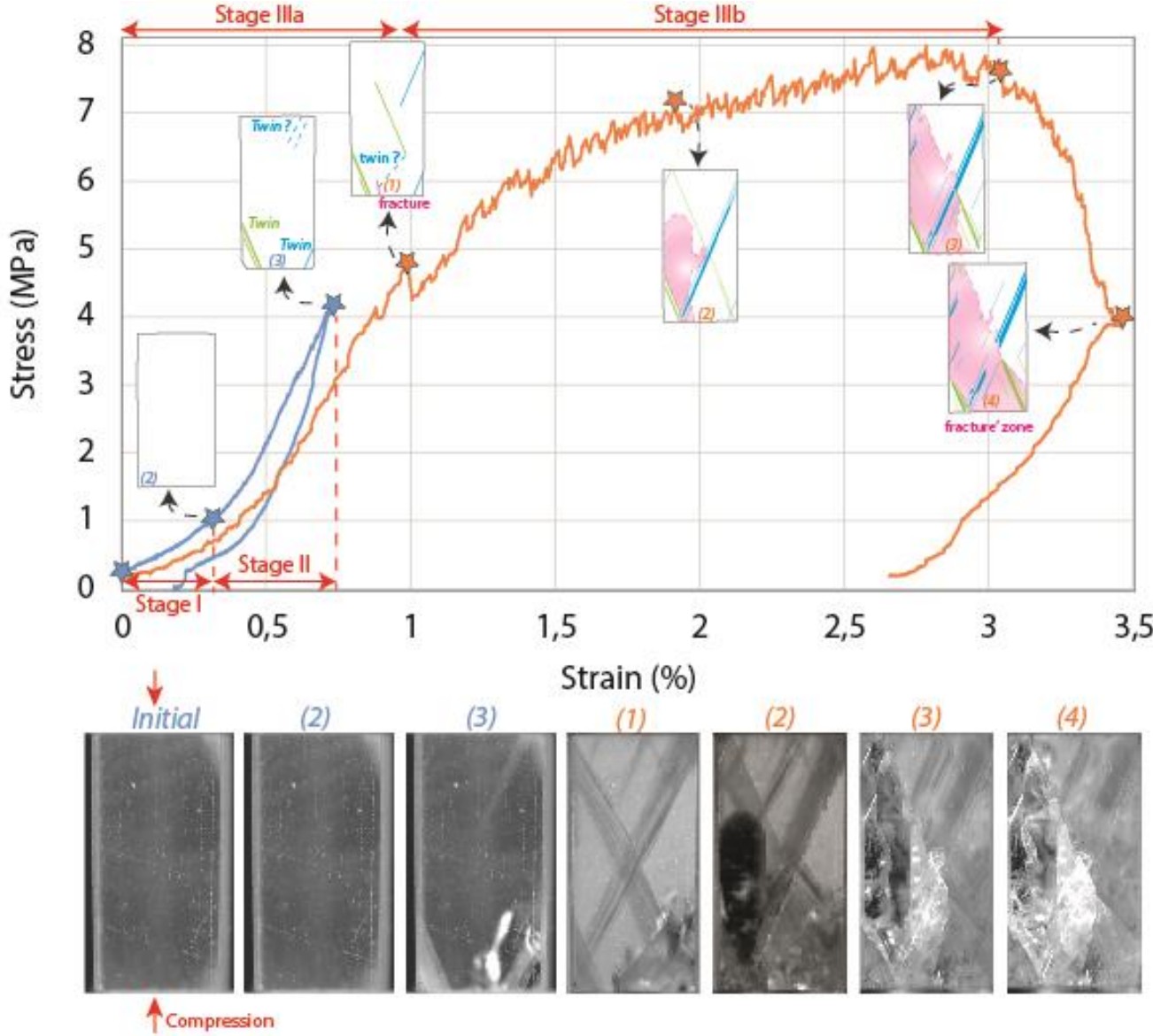

**Figure 4. Results from sample 4 x 4 x 8 mm3 compressed along the direction $[2\bar{1}\bar{1}0]$. Same key as in Fig.2.**

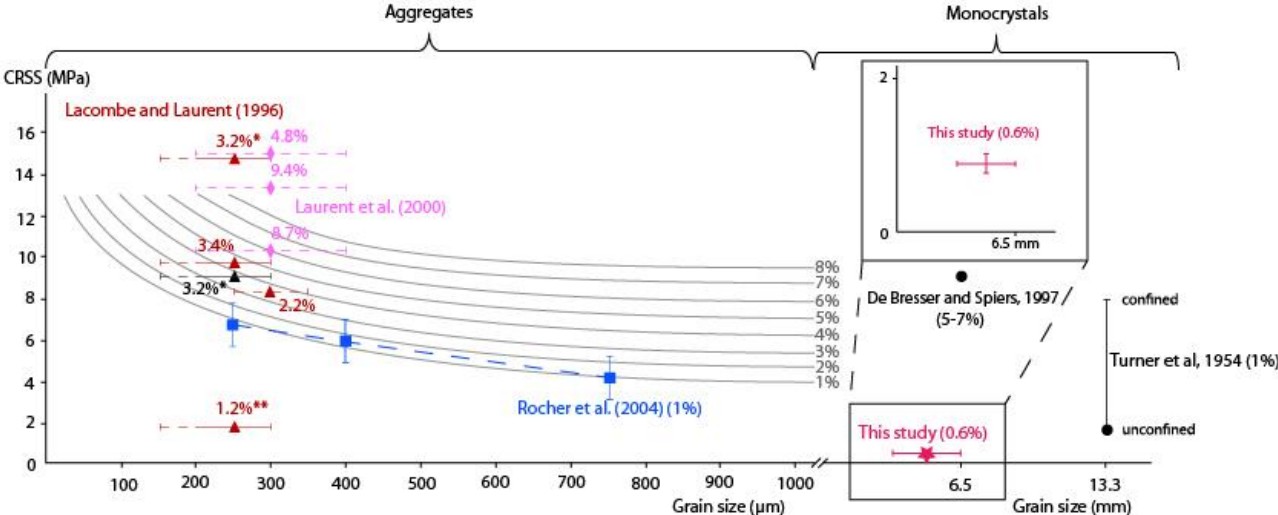

**Figure 5. Summary of the current knowledge about the evolution of the CRSS value as a function of grain size and strain. The figure has been modified from an initial drawing by Amrouch (2010) on the basis of data by Rocher et al. (2004). Constraints from the earlier and present studies are reported (De Bresser and Spiers, 1997; Lacombe and Laurent, 1996; Laurent et al., 2000; Turner et al., 1954). (\*) Initial value from Lacombe and Laurent (1996) re-evaluated by Rocher (1999). (\*\*) Outliers due to data issues.**