# Peer review of "Uniaxial compression of calcite single crystals at room temperature: insights into twinning activation and development"

_Solid Earth, 2018_

## Referee Comment (RC1) · H. de Bresser (Referee) · 19 Oct 2018

The idea behind the set of experiments reported in this paper is good: previous experiments on calcite twinning have almost always been done at high temperature, bringing up questions about the extrapolation to the lower temperatures of importance in calcite deformation in nature.

De experiments as such are well described, and the approach with high-resolution pictures made during the experiments is of great value compares with a post-mortem approach.

[Figure]

However, I find the number of samples used, namely 3, really too low to make conclusions with confidence about the effect of orientation and the effect of grain size. One would at least want to see duplication tests. But prefarably more

For example, CRSS values obtained from the tests were 0.44 MAP in experiment 1 ([2110] orientation), and about 1.2 MPA ([0110] orientation, two sizes). That is roughly a factor of 3 difference. That is a rather large uncertainty. Many other CRSS determinations have shown higher values (see Fig. 5). The CRSS in the current paper are based on taking the overall stress value at the moment of the first twin, but perhaps there are local stress concentrations that play a role. The different strains at which the first twin develops (0.3-0.7%) and the fact that microcracks develop already at low strains might be an indication for such stress concentrations. The authors do bring up this point in the discussion (p.7), but without a clear inference.

I would say that these first results are promising, but that more experimental work is needed to get a convincing story.

One other question I have is: are you sure that no other mnechanism than twinning plays a role in straining? Does the volume of twins fit the amount of axial strain?

And finally, a couple of very good points are mentioned as discusison points, but the paper doesn't really go into it in-depth. I actually think it should. It concerns: - the possible role of (the lack of) confining pressure; relevant when extrapolating to nature. - the effect of multiple grain boundaries in an aggregate, i.o.w. how to relate single crystal observations to polycrystals.

I do want to motivate the authors to continue the project. At present the basis is a bit thin.

In terms of presentation: it would be useful to clearly indicate the stages I, II, IIIa en IIIb in Figures 2, 3 and 4.

---

## Referee Comment (RC2) · Anonymous Referee #2 · 12 Dec 2018

Dear editor,

I read the manuscript submitted for publication in Journal Solid Earth and entitled "Uniaxial compression of calcite single crystals at room temperature: insights into twinning activation and development".

The authors present a new experimental study on the nucleation and development of twinning in single calcite crystals subjected to uniaxial compression stress at room temperature. While the samples are being deformed at constant strain rate, optical and electronical pictures are also taken. The paper builds on the analysis of the mechanical data and pictures produced throughout the deformation of the samples. I enjoyed

reading the paper as it provides interesting new data and associated discussion. It also raises interesting questions. However, I have some comments that the authors may want to consider.

In this study, single crystals were used to get rid of the influence of grain boundaries and size distribution on the development of twinning as highlighted by Newman (1994). This is an interesting idea to me but I am a bit disappointed by the fact that the authors do not discuss this point thoroughly in the manuscript. A question I still have is how can these results be extrapolated to natural conditions in which grain size distribution is obviously much more complex? Starting with the simplest situation seems to be an interesting idea but they could maybe discuss in more details the implications of this (over?-)simplification.

As clearly stated by the authors, twinning activation is mainly dependent on differential stress and grain size. In this study, 4x4x8 mm3 and 3x3x6mm3 single crystals were used, which are much larger than most crystals found in natural rocks. As the authors used two different grain sizes, I thought that they could discuss this influence of grain size on the development of twinning. This point is slightly discussed in section 3.2.4 in which the authors compared their results to previous studies. Why not compare their own different experiments first? Side question: their results seem to compare to previous results obtained by Turner et al (1954) on unconfined samples: Do the authors think that confinement might play a role on the development of twinning, contrary to what is claimed in the introduction? One idea might be that twinning is actually slightly associated with microcracking which is highly dependent on confining pressure.

The results obtained in this study would allow the authors to draw the evolution of total twin thickness as a function of axial strain. Did they have a look at this correlation? It may help them decipher whether twinning is associated with microcracking or not (and even maybe try to quantify strain due to each of these two micromechanisms if they are associated).

Interestingly, the authors mention that the duration of stress application has a great impact on twin lamellae thickness (line 30, page 6). To me, this may imply that making a creep experiment would be of interest, also since natural conditions may be closer to constant stress conditions rather than constant strain rate deformation.

Finally, stress-strain curves show multiple small stress drops. Are these stress drops associated with microcracking or twin nucleation or not?

As a conclusion, these experimental results seem to be interesting to me but I think that the authors could go deeper in-depth in their conclusions, maybe after performing a few additional experiments.

---

## Author Comment (AC1) · 12 Dec 2018

First of all, we would like to thank you for the time you have taken to read and write down your comments about this article.

First comment: "I find the number of samples used, namely 3, really too low to make conclusions with confidence about the effect of orientation and the effect of grain size. One would at least want to see duplication tests. But prefarably more" Actually, it was not only 3 samples that have been tested in this study. We have chosen to present only the results of 3 representative ones, because we find it clearer for the main message. But we did test 4 synthetic "perfect samples" (purchased from SurfaceNet supplier).

The respective loading curves provided in the paper (and now in the supplementary material) show that the results are qualitatively (and quantitatively) reproducible with respect to the different stages characteristic of the elastic and plastic phases. Two samples have dimensions of 3x3x6 mm3 and 2 other samples of 4x4x8 mm3. For both sizes there are two different lattice orientations, which are chosen in order to favour twinning. In addition to these samples, we also used several specimens cut from optically clear Iceland Spaths. They were used to establish the appropriate deformation protocol to be applied to the synthetic strain-free samples, in terms of sample preparation, loading rates, and conditions for in the situ optical/SEM monitoring. This preparation phase allowed for instance to select the most suitable CCD cameras and to establish the SEM imaging conditions. Unfortunately, because of the natural origin of the samples pre-existing cleavage cracks, twin lamellae and micro-fluid inclusions preclude the extraction of perfectly similar samples with respect to size, crystal orientation and pre-existing defects which can cause local stress concentration and the possibility to precisely infer the CRSS value (Critical Resolved Shear Stress) of interest. Hence, Iceland Spaths are not appropriate for this study, but only for training. However, interestingly the corresponding results showed qualitatively the same patterns for the loading curves as for the "perfect samples", which further demonstrates the reproducibility of the results. A supplementary data has now been added to this article to show the results of some of these preliminary tests on Iceland Spaths, as well as the result for the fourth "perfect sample". Concerning the relationship between the grain size and the CRSS value, as said in the discussion part and showed in the Figure 5 of the article, no reliable conclusion can be drawn, and we discuss the reasons why. We did not report any link between the lattice orientation and the CRSS value.

Changes have been made in the text between line 30 page 2 until line 5 page 3, line 11 to 12 page 3 and line 4-6 page 5 + information has been added about the camera and software used line 18 page 3.

Second comment: "The CRSS in the current paper are based on taking the overall

stress value at the moment of the first twin, but perhaps there are local stress concentrations that play a role. The different strains at which the first twin develops (0.3-0.7%) and the fact that microcracks develop already at low strains might be an indication for such stress concentrations. The authors do bring up this point in the discussion (p.7), but without a clear inference." We have chosen to work on synthetic material precisely in order to guarantee the quality of the single-crystal: limited dislocation density, fluid inclusion-, cleavage- and twin-free, which would undoubtedly influence the calculated CRSS through inherited work hardened state and local potential stress concentrations. We disagree with what the reviewer wrote about the appearance of the first twins at different strains. The total strains considered by the referee (0.3 – 0.7%) are only apparent values, which are not necessarily representative of the plastic yielding, because there is an important emplacement phase (stage I) during which the full contact between pistons and samples is progressively established. The latter is not the same for each specimen, because it depends on the exact sample geometry (parallelism of loaded faces).Moreover, eventual stress concentrations would expectedly be more pronounced during this emplacement stage, however the micro-cracks did not appear at this stage. Micro-cracks essentially developed latter on, during the pseudo-elastic and plastic phases. This may indicate that micro-cracking is a complementary and necessary deformation mechanism, allowing for accommodation of local incompatibilities of strain, for instance at sample-piston interfaces when contact have been made (stage II and III). The latter are actually expected, because twinning is not isotropic and the pistons have not enough degrees of freedom to fit the inhomogeneous sample strain if solely resulting from twinning. Conversely, it is in practice impossible to completely prevent local stress concentration near the crystals ends, due to the piston-sample frictional contacts. The early fractures are likely due to the friction between the sample and the pistons and the fact that the loading geometry is axial, but twinning related strain is not axi-symmetric. The calcite crystal lattice stereo-diagram (Figure 1 in the main text) show that the deformation systems are not symmetrical along the stress axes. Therefore, when only twinning deformation mechanism is activated the deformation cannot be axisymmetric. Twinning activation leads to the formation of shear bands composed of several twin lamellae which result in a shortening component along the stress axis, but also in extensional transverse components. The latter are perpendicular to the stress orientation, but are not equivalent (not axisymmetric). However, the pistons remain aligned and cannot accommodate a later motion during the shearing of the crystal imposed by twinning. Similarly to uniaxial compression of a sample activating crystal slip onto a single slip system, the sample must either rotate, or the pistons must follow the lateral motion. If impossible, there occurs frictions and stress concentrations at the sample – piston interface, with possibly local micro-fracturing. The fact we use samples with a length of twice the width (3 x 3 x 6 and 4 x 4 x 8 mm3) allows however to keep the central part of the crystal free of these edge effects, so one can safely consider that our results and inference are reliable.

Information have been added in the main text between line 10 and 17 of the page 6.

Third comment: "One other question I have is: are you sure that no other mechanism than twinning plays a role in straining? Does the volume of twins fit the amount of axial strain?" We do not have any monitoring of what happens in the inner part of the crystal but only on one surface. Fortunately however, it is possible to have a reliable guess of what is going on within the crystal by transparency (pure calcite crystals are transparent). There is no optically observable deformation mechanism other than fracturing and twinning, but since the tests are performed at room temperature and at relatively high strain rate we do not expect any noticeable crystal slip plasticity. Once fracturing impacts a large portion of the crystal it is difficult to quantify the partitioning of the shortening along the stress axis due respectively to fracturing and twinning. Therefore, we measured the shortening after the EBSD analysis performed between the two consecutive loading phases. For the example of the crystal of 3 x 3 x 6 mm3 compressed along [21 ÌĚ1 ÌĚ0], the whole shortening after the first loading was of about 151 $\mu$m. High resolution SEM micrographs were used to measure twin lamellae thicknesses, which allows to determine that twin is responsible of the axial deformation for 116 $\mu$m.

Optically we can see by transparency a few twin lamellae which are not crossing the whole crystal to the surface and this can explain for part of the difference we calculate with the prescribed shortening. The optical error of measurement of the individual twin lamellae thicknesses on pictures may also contribute to the difference.

Information has been added line 17 to 22 of the page 4.

Fourth comment: "the possible role of (the lack of) confining pressure; relevant when extrapolating to nature." The lack of confining pressure probably leads to easier micro-fracturing (with some contribution to the whole deformation), but it does not affect the differential stress required to activate twinning. But the main issue for extrapolation of the data collected from single crystals to natural aggregates resides in having free surfaces instead of neighbour grains, which presence necessarily modifies the local stress state. The problem consists in extrapolating the uniaxial state to a grain within a polycrystal (see next question too). The twin lamellae probably thicken up to 90 $\mu$m wide also due to the free surfaces. So we acknowledge that the results from single-crystal experiment cannot be extrapolated directly and easily to natural aggregates.

Information has been added line 20 to 22 of the page 7.

Fifth comment: "the effect of multiple grain boundaries in an aggregate, i.o.w. how to relate single crystal observations to polycrystals." The choice of single crystals instead of aggregates is motivated by the aim of obtaining the purely intrinsic, material depen-dent CRSS value of the twin systems. For that, we need the actual stress state on the considered crystal, which can only be achieved with single crystals. The referee is perfectly aware of the problem, since he has himself extensively worked with the same methodology on crystal slip plasticity of calcite. The application of mechanical data gathered on single crystals to a polycrystal is another topic. It is known for instance that grain boundaries act as barriers for the propagation of twins (Barber and Wenk, 1979) and that some additional deformation mechanisms (pressure-solution, . . .) are required to maintain geometric and kinematic consistency between grain within the

aggregates and/or to release elastic strain due to these local geometrical incompatibilities. For that purpose, analytical mean field, or computational full field homogenization models must be applied but this is much beyond the scope of the present work.

Information has been added line 26 to 33 of the page 8.

Sixth comment: "In terms of presentation: it would be useful to clearly indicate the stages I, II, IIIa en IIIb in Figures 2, 3 and 4."

Done as suggested.

Please also note the supplement to this comment:
https://www.solid-earth-discuss.net/se-2018-80/se-2018-80-AC1-supplement.zip

―――――――――――――――――

---

## Editor Comment (EC1) · Heap (Editor) · 13 Dec 2018

Dear authors,

As you can see, I have now received two reviews of your manuscript. Both reviews have a positive tone. "The idea behind the set of experiments reported in this paper is good...the approach with high-resolution pictures made during the experiments is of great value..." (Reviewer #1). "I enjoyed reading the paper as it provides interesting new data and associated discussion" (Reviewer #2). However, the reviews also highlight areas in which the manuscript can be improved. For example, both reviewers ask for clarification on the role of confining pressure, whether there are other micromechanisms at play, and whether or not the authors should consider running additional experiments to bolster the discussion topics. If you are willing, please now prepare a point-by-point rebuttal letter and revised manuscript.

Thank you for choosing Solid Earth. We hope that you consider us again for your future manuscripts.

Mike Heap (Topical Editor of Solid Earth)

———————————————————

---

## Author Comment (AC2) · 4 Jan 2019

Author's answers to the interactive comment of Anonymous (referee) Thank you for the time you spent on reading and writing comments about this article.

First comment: "A question I still have is how can these results be extrapolated to natural conditions in which grain size distribution is obviously much more complex? Starting with the simplest situation seems to be an interesting idea but they could maybe discuss in more details the implications of this (over?-)simplification." The single-crystal approach does not intend to be directly applied to natural aggregates. It follows a classical methodology in determining the intrinsic mechanical properties (elastic moduli,

thermal expansion, flow strength, resolved shear stresses, anisotropy...) of minerals and crystalline materials. Such fundamental data are the basic input for homogenization models (such for example, mean field elasto-visco-plastic self-consistent schemes or full field fast Fourier transform ones), which intend to calculate the average mechanical properties of aggregates and rocks. The aggregate behaviour is the final target, and it largely depends on phase proportions and microstructures, such as spatial distribution of phases and porosity, crystallographic texture..., which complexity has to be specifically addressed in the homogenization type of model. As you highlighted in your review it has to take into account the grain size (and its distribution), texture, porosity, etc... which impact the local interactions and dictate each grain boundary conditions (deformation compatibility between grains and local stress state resulting from these grain interactions). So, we do not pretend there is simple direct extrapolation of measurements performed on single-crystal to aggregates. However, our aim is specifically to provide the critical resolved shear stress (CRSS) value needed to activate the twinning phenomenon. The result can be considered as the basic input for further homogenization modelling. Most importantly, it is an intrinsic crystal related parameter, which is not depending on microstructure of the aggregates. From the point of view of the CRSS value, the differential stress is the prime variable, whilst confining pressure (and possibly temperature) is a second order one. The role of the grain size, which is still being debated, enters in the category of microstructure parameters, important to be considered for aggregates, but not for single crystals.

Information have been added in the main text line 9-12 page 9 + the review from the previous reviewer line 26-27 page 9, line 31 page 9 to line 2 page 10.

Second comment: "Why not compare their own different experiments first? Side question: their results seem to compare to previous results obtained by Turner et al (1954) on unconfined samples: Do the authors think that confinement might play a role on the development of twinning, contrary to what is claimed in the introduction? One idea might be that twinning is actually slightly associated with microcracking which is

highly dependent on confining pressure." According to Covey-Crump et al., (2017) the CRSS value is supposed to follow the Hall-Petch equation, with power law decreasing of CRSS for increasing grain size. But, as shown in the figure 5 of the main text, the crystal sizes of our experiments correspond to the asymptotic part of the curves, where experimental uncertainties may obscure the possibly small differences of CRSS with respect to the small differences in crystal sizes. Considering only our results did not show any clear trend in CRSS evolution, with a mean value of 0.90 MPa and a standard deviation of 0.35 MPa, so that we do not extend further the discussion in comparing only our data. Comparing our data with those from other similar studies is not easy as well: - Turner et al. (1954) used natural Iceland spaths. We show in the complementary data that we also used Island spaths for establishing the experimental protocol. The results are not directly comparable with those obtained from the synthetic (optical quality) crystals. As said in the main text, the natural single-crystals of calcite are already slightly strained (some twins are visible), or cleaved, and most contain micro-fluid inclusions. All these imperfections induce strain hardening, and hence increased CRSS values for twinning activation. (see fig. 2, 3 and 4 as in the supplementary data added after the first review). This is why our, synthetic samples provide a CRSS value slightly lower than that of Turner et al. (1954) (as expected). Using natural samples implies undefined initial state, which questions the validity of the retrieved CRSS values, when applying inversion techniques in order to determine paleo differential stresses. - De Bresser and Spiers (1997) annealed their Iceland spath specimens in order to restore the previous cumulated strain. Unfortunately, as explained in the main text, this procedure leads to the formation of sub-grain boundaries, that affect the propagation and spreading of twins and cause non-uniform stress distribution at the grain scale, which is potentially biasing the results. Also for both of these previous studies, the CRSS determination was based on post mortem observations, which precluded accounting for the loading history and probably the detection of the earliest twinning events. The lack of confinement is certainly responsible to some extent of micro-cracking. Though, micro-cracking is probably unavoidable in compression loading geometry, with laterally

bound pistons. Indeed, twinning is anisotropic and strain incompatibilities must arise at the sample pistons interfaces. The latter would necessarily result in crystal rotation and frictions in the vicinity of the interface. Micro-cracking could actually be a necessary local accommodation mechanism. Conversely, confining pressure is unlikely to be directly involved in twinning activation, which is mostly dependent on differential stress. However, we must admit that confining the samples with a fluid, or leaving an unconfined free surface during uniaxial compression could affect twin thickening, with respect to confinement by crystalline grains. In the latter case, within a polycrystal, the crystallographic orientation of the neighbours will greatly condition the amount of crystal shear that can be transmitted across the grain boundary. The crystallographic constraints, in terms of shear strain compatibility, along the grain boundary would favour the occurrence of numerous distributed thin twins. Conversely, the presence of fluid confined free surface could favour the localized development of thick twins.

Information have been added in the main text line 26 page 7 to line 2 page 8 and 14-29 page 8.

Third comment: "The results obtained in this study would allow the authors to draw the evolution of total twin thickness as a function of axial strain. Did they have a look at this correlation? It may help them decipher whether twinning is associated with microcracking or not (and even maybe try to quantify strain due to each of these two micromechanisms if they are associated)." Unfortunately, it is not possible to simultaneously monitor both the scale of the sample and the scale of individual twin lamellae. The whole sample surface is observed by optical microscopy in order to detect where and how many twins occur. But, optical resolution does not allow to follow very precisely the thickening of the latter. In order to be able to follow the thickening of each twin lamellae we would have been obliged to dismount the specimens from the loading stage for closer observations in the SEM. However, such a step-by-step approach would have induced cyclic loading, with a priori unknown effects on twin activity, but with serious risks of enhancing micro-fracturing. Besides, each emplacement on the

loading stage could also modify the boundary conditions at piston-sample interfaces.

Fourth comment: "Interestingly, the authors mention that the duration of stress application has a great impact on twin lamellae thickness (line 30, page 6). To me, this may imply that making a creep experiment would be of interest, also since natural conditions may be closer to constant stress conditions rather than constant strain rate deformation." We are actually not aware of any creep experiment where the macroscopic flow is ensured solely by twinning. The experiment is certainly interesting to do. But, it is clearly not the purpose of this work. On the other hand, it is not clear if constant stress or strain rate is more representative of natural conditions. It might actually depend on the geodynamical context.

Fifth comment: "Finally, stress-strain curves show multiple small stress drops. Are these stress drops associated with microcracking or twin nucleation or not?." We are absolutely clear about the fact that the stress drops are associated with twinning. Unfortunately, it is very difficult to draw a figure showing the entire loading curve, but that is also able to clearly highlight the fact that the stress drops relate to twinning. However, it is clear that the micro-cracking events are not numerous enough to account for all the serrations (stress drops). Besides, the first loading curve corresponds to a crack-free sample and still shows the dense serrations.

Please also note the supplement to this comment:
https://www.solid-earth-discuss.net/se-2018-80/se-2018-80-AC2-supplement.zip